# Selecting the Best Combined Biological Therapy for Refractory Inflammatory Bowel Disease Patients

**DOI:** 10.3390/jcm11041076

**Published:** 2022-02-18

**Authors:** Eduard Brunet Mas, Xavier Calvet Calvo

**Affiliations:** 1Servei Aparell Digestiu, Hospital Universitari Parc Taulí, 08208 Sabadell, Spain; ebrunetm@tauli.cat; 2Departament de Medicina, Universitat Autònoma de Barcelona, 08193 Bellaterra, Spain; 3CIBERehd, Instituto de Salud Carlos III, 28029 Madrid, Spain

**Keywords:** inflammatory bowel diseases, biologic treatment, combination, Crohn’s disease, ulcerative colitis

## Abstract

Current medical treatment for inflammatory bowel disease (IBD) does not achieve 100% response rates, and a subset of refractory and severely ill patients have persistent active disease after being treated with all possible drug alternatives. The combination of two biological therapies (CoT) seems a reasonable alternative, and has been increasingly tested in very difficult cases. The present review suggests that CoT seems to be safe and effective for refractory and severely ill IBD patients. Ustekinumab plus vedolizumab and vedolizumab plus anti-TNF were the most used CoTs for Crohn’s disease. For ulcerative colitis, the most used CoTs were vedolizumab plus anti-TNF and vedolizumab plus tofacitinib. The aforesaid CoTs have shown good efficacy and few adverse events have been reported.

## 1. Introduction

Currently, there is a reasonable number of useful therapies for IBD. The treatment armamentarium includes small molecules and biological treatments. Small molecules include classical drugs such as mesalazine, corticosteroids, and immunosuppressant treatments; this latter group includes thiopurines, methotrexate for Crohn’s disease (CD), and tofacitinib for ulcerative colitis (UC). Regarding biological treatments, anti-TNF drugs, vedolizumab, and ustekinumab are currently widely used [1,2].

These drugs are mostly used sequentially. Thus, if a drug is ineffective in controlling IBD symptoms, that drug is withdrawn and replaced by another (for example, in patients who have active disease despite receiving an anti-TNF drug, treatment with the anti-TNF drug would be stopped and the patient would begin treatment with an alternative biological drug). The only exception is the combination of an anti-TNF plus azathioprine, which has been widely used in clinical practice since the 2010 SONIC trial showed that this combination achieved higher remission and mucosal healing rates than monotherapy without a clear increase in adverse events [3].

Used individually, IBD therapies reach a maximum clinical remission rate of approximately 40–60% [4]. Therefore, current medical treatment for IBD does not achieve 100% response rates, and a subset of refractory and severely ill patients have persistent active disease after being treated with all the possible drug alternatives. These patients often require aggressive rescue therapies such as major surgery or bone marrow auto-transplant in CD, or proctocolectomy in UC [5,6].

As these rescue treatments have significant risks and may have a negative impact on quality of life, the combination of two biological therapies (CoT) seems a reasonable alternative. In fact, CoT has been increasingly tested in very difficult cases and in two clearly different settings: in patients with uncontrolled IBD, and in patients with controlled IBD but extraintestinal manifestations that did not respond to a single biological therapy [7]. The safety and efficacy of CoT have mostly been reported as case reports or short series [4].

Two meta-analyses and one large review on CoT have been published to date [7,8,9]. Neither of them reported the efficacy and safety of the individual CoT evaluated.

After the publication of these articles, a large case series study gathering cases from many European centers was published (104 combinations in 98 patients (75 for IBD and 23 for uncontrolled extraintestinal manifestations)). This study, along with a few other case series, reported data separately for the most used CoT, allowing a first attempt in pooling results to evaluate their individual safety and efficacy [10].

The aim of this review was to describe the data on the effectiveness and safety of the most popular biological CoT for refractory IBD patients. Individual case reports and pediatric studies were not included in the review.

## 2. Global Efficacy and Safety of CoT

Two meta-analyses and two large reviews have been published to date:

In an early study, Ribaldone et al. reviewed seven studies (18 patients) with a combination of TNF inhibitors and vedolizumab as well as vedolizumab and ustekinumab. Clinical improvement was seen in all patients, and endoscopic improvement was reported in 93% of patients. No safety concerns were identified [9].

Ahmed et al. in a recent meta-analysis reviewed 30 studies reporting 288 trials of dual biologic or small-molecule therapy in 279 patients. The most common CoT was anti-TNF and vedolizumab (48%). The pooled clinical remission was 59% (95%CI 42–74%) and the endoscopic remission was 34% (95%CI 23–46). They observed 31% (95%CI 13–54%) of adverse events, but only 7% (95%CI 2–13%) were severe or life-threatening [8].

In 2021, Gold et al. published a review pooling data from 209 CoTs. They included retrospective studies, case reports, and case series. This review suggested that dual biologic therapy may be effective at inducing remission in patients with refractory luminal symptoms and/or extraintestinal manifestations. They reported an efficacy ranging from 67% to 80%. No severe adverse events were described [7].

After adding a large recent European study [10] to the previous studies, anti-TNF plus vedolizumab and vedolizumab plus ustekinumab emerged as the most used CoTs. Less-frequent combinations included anti-TNF plus ustekinumab and anti-TNF plus tofacitinib, with the latter mostly used in UC.

## 3. Usefulness and Safety of Biologic Combinations

Table 1 shows the studies on the use of combination targeted therapy in IBD in the adult populations included in the review. Additionally, Figure 1 shows the pooled rates of clinical response, clinical remission endoscopic response, endoscopic remission, and adverse event rates for the most used CoTs.

(a)Ustekinumab plus vedolizumab

Yang et al. [11] reported the results of eight patients who received treatment with the combination of ustekinumab and vedolizumab. During follow-up at week 40, five of seven (71%) patients achieved clinical response, four of seven (57%) achieved clinical remission, five of eight (63%) achieved endoscopic improvement, and two of eight (25%) achieved endoscopic remission. The adverse event rate was low—one of eight (13%) patients.

Kwapisz et al. [12] reported the results in five patients with this CoT. Four of five (80%) had a clinical response, and no adverse events were reported.

Privitera et al. [13] reported results in three patients. All (100%) had a clinical response but none had a clinical remission (0%) at 6 months of follow-up. One patient (33.3%) presented a perianal abscess as an adverse event.

In the US series of Glassner et al. [14], 25 patients received ustekinumab and vedolizumab. Unfortunately, individual results were not available.

Finally, in the European study of Goessens et al. [10], 21 patients received this CoT. Endoscopic response was observed in 11 of the 13 patients with CD evaluated (85%) after an 11-month follow-up.

(b)Anti-TNF plus vedolizumab

Yang and colleagues [11] reported the results of 12 patients who received CoT with anti-TNF and vedolizumab. During follow-up at week 40, 5 of 12 (42%) patients achieved clinical response, 4 of 12 (33%) achieved clinical remission,4 of 12 (33%) achieved endoscopic improvement, and 3 of 12 (25%) endoscopic remission. The adverse event rate was low (2 of 12 (15%)).

Kwapisz et al. [12] reported results in eight patients. Five of eight (62.5%) had a clinical response. Three of eight patients (37.5%) presented infections as an adverse event.

Privitera et al. [13] reported results in six patients. Three of six (50%) patients had a clinical response and three of six had clinical remission (50%) at 6 month follow-up. Only one adverse event was reported in one patient (16.6%), who presented a cutaneous rash.

Glassner et al. [14] included seven patients on anti-TNF plus vedolizumab in their US series, however, individual results were not available.

Finally, Goessens et al. [10] reported 41 patients with this CoT. The endoscopic response was observed in 16 of the 25 (64%) patients with CD and in 8 of 11 (67%) patients with UC evaluated at 11 months of follow-up.

(c)Other combinations

Yang and colleagues [11] reported the results of three patients who received treatment with the combination of anti-TNF and ustekinumab. During follow-up at week 40, one of three (33%) patients achieved clinical response, one of three (33%) achieved clinical remission, one of three (33%) achieved endoscopic improvement, and one of three (33%) achieved endoscopic remission. No adverse events were observed.

Kwapisz et al. [12] reported the results in two patients on anti-TNF and ustekinumab. Both patients (100%) had a clinical response and neither patient presented an infection as an adverse event.

Privitera et al. [13] reported results in four patients on anti-TNF plus ustekinumab; one patient had a clinical response and three had clinical remission at 6 month follow-up.

Glassner et al. [14] reported eight patients on vedolizumab plus tofacitinib, nine patients on anti-TNF plus tofacitinib, and three patients on tofacitinib plus ustekinumab. However, individual results were not available.

Finally, Goessens et al. [10] reported on 12 UC patients treated with vedolizumab plus tofacitinib. Endoscopic response was observed in 8 of the 12 patients (67%) evaluated after an 11 month follow-up. Other combinations were anti-TNF plus ustekinumab in 11 patients and tofacitinib plus anti-TNF in 1 patient.

(d)Safety

Although the studies did not give individual data of each combination, they do offer an overview of the safety of combination treatment. In the European study of Goessens et al. [10], 42 of 98 (42%) patients experienced a total of 42 significant adverse events. Serious opportunistic infections occurred in 10 of 98 patients, 6 in the group of anti-TNF plus vedolizumab, 3 with anti-TNF plus ustekinumab, and 1 with ustekinumab plus vedolizumab. All of them resolved. Life-threatening adverse events were observed in two patients (angioedema and hypersensitivity to infliximab) [10]. In the American study, 13 patients (26%) experienced 23 adverse events; 8 were serious infections (1 bacterial enteric infection, 3 postoperative infections, 2 pelvic abscesses, and 2 infections of intravenous catheters) and the remaining 15 adverse events were mild (7 enteric infections, 7 pulmonary infections, 3 postoperative infections, 1 viral wart, 1 urinary tract infections, 2 pelvic abscesses, and 2 catheter infections) [14]. Table 2 shows the rates of clinical response, clinical remission, endoscopic response, endoscopic remission, and adverse event rates for each study and each CoT.

## 4. Discussion

Although data are still preliminary, CoT may quickly become a must for IBD specialists. Uncertainty remains, but first reports suggest more than reasonable efficacy and safety for very severely ill IBD patients. Even though CoT is not 100% efficacious and may carry significant adverse events, risks and benefits should be balanced with those of the current treatment of uncontrolled severe IBD (multiple surgeries, proctocolectomy with ileostomy, ileoanal reservoir, or autologous bone marrow transplantation). So, it seems reasonable that, after a careful discussion of potential risks and benefits, most patients will opt for a CoT trial before progressing to more aggressive approaches.

From our data, it is not possible to give a clear recommendation of which combination should be used. The most-used CoTs are shown in Table 3. Of them, vedolizumab plus ustekinumab and vedolizumab plus anti-TNF were the most effective CoTs for CD. Furthermore, vedolizumab plus anti-TNF and vedolizumab plus tofacitinib were the most effective CoTs for UC. The combination of ustekinumab and vedolizumab seems especially attractive because it might combine efficacy, safety, and persistence over time. Very recently, Stone et al. reported similarly good results in a retrospective series of 10 patients. However, data are currently incomplete as the study has been published only as an abstract [15]. Data are preliminary and, in patients with UC and uncontrolled extraintestinal manifestations, CoT including anti-TNF or tofacitinib might be more effective.

CoT has also been explored in other clinical settings, such as the treatment of psoriasis with associated joint manifestations. In these patients, treatment was effective and there was no increase in adverse events [16,17,18]. Otherwise, the combinations used for the treatment of rheumatoid arthritis demonstrated good efficacy but an increase in the rate of adverse events [19,20,21,22]. In IBD, CoT has even been used in pediatric patients with good results and safety [23,24].

There are multiple limitations of this review. The most important is the low number of cases to date and the high risk of selection bias. In this sense, trials with good results are likely more reported than those without efficacy or with severe adverse events. Additionally, data on the safety of each combination are currently lacking. However, the largest series [10,14] reported a low global rate of adverse events, suggesting that most individual combinations may be safe. Particularly in the study of Goessens et al. [10], safety results were analyzed globally and patients with CoT for extraintestinal manifestations cannot be excluded.

Prospects for combination therapy are multiple. For example, as mucosal healing has been shown to be an extremely good prognostic factor [25,26], initial CoT aimed to achieve early mucosal healing may have the potential to modify the natural history of IBD. However, in our opinion, prudence should be applied. Careful evaluation of CoT in a multidisciplinary committee before approval might further enhance both patient and doctor safety. Furthermore, the patient needs to be clearly informed about the benefits and risks of CoT. Finally, we recommend that informed consent be obtained for any CoT trial.

In conclusion, IBD treatment is still rapidly evolving. Along with the new therapies that are rapidly becoming available, CoT has demonstrated promising results and may represent a new opportunity to improve both patients’ quality of life and long-term prognosis. However, current data are very limited, and larger studies with longer follow-up are desirable to confirm the safety and efficacy of CoT. In the meantime, CoT seems a real alternative for refractory and severely ill patients who cannot wait for new developments to come.

## Figures and Tables

**Figure 1 jcm-11-01076-f001:**
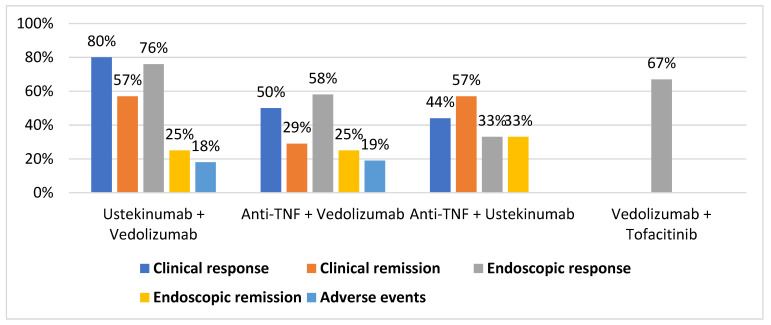
Percentage of clinical response, clinical remission, endoscopic response, endoscopic remission, and adverse events for each combination therapy.

**Table 1 jcm-11-01076-t001:** Data on the use of combination targeted therapy in IBD in adult populations.

Reference	Year	Study Type	Number of Subjects	Disease	Age (Mean)	Disease Duration (Mean Years)	Clinical Evaluation	Endoscopic Evaluation	Adverse Events	Follow-Up (Mean)
Goessens et al. [10]	2021	Multicentric Retrospective	98	58 CD 40 UC	26		70% response	50% response	42%	8 month
Glassner et al. [14]	2020	Unicentric Retrospective	50	32 CD 18 UC 1 IBD-U	36.7	14.8	50% remission	34% remission	16%	8 month
Kwapisz et al. [12]	2021	Unicentric Retrospective	15	14 CD 1 UC	36	12.5	73% response	44% response	53%	24 month
Privitera et al. [13]	2020	Multicentric Retrospective	16	11 CD 5 UC	38	10.5	100% response		18.8%	7 month
Yang et al. [11]	2020	Multicentric Retrospective	22	22 CD	35		50% response	50% response	13%	9 month

**Table 2 jcm-11-01076-t002:** Results for clinical response, clinical remission, endoscopic response, endoscopic remission, and adverse events for each combination and each study.

		Clinical Response	Clinical Remission	Endoscopic Response	Endoscopic Remission	Adverse Events
**Ustekinumab + Vedolizumab**	Yang et al.	5 of 7	4 of 7	5 of 8	2 of 8	1 of 8
	Kwapisz et al.	4 of 5				0 of 5
	Privitera et al.	3 of 3				1 of 3
	Glassner et al.			11 of 13		
	**TOTAL**	12 of 15 (80%)	4 of 7 (57%)	16 of 21 (76%)	2 of 8 (25%)	2 of 11 (18%)
**Anti-TNF + Vedolizumab**	Yang et al.	5 of 12	4 of 12	4 of 12	3 of 12	2 of 12
	Kwapisz et al.	5 of 8				3 of 8
	Privitera et al.	3 of 6	3 of 6			1 of 6
	Glassner et al.			24 of 36		
	**TOTAL**	13 of 26 (50%)	7 of 18 (29%)	28 of 48 (58%)	3 of 12 (25%)	5 of 26 (19%)
**Anti-TNF + Ustekinumab**	Yang et al.	1 of 3	1 of 3	1 of 3	1 of 3	
	Kwapisz et al.	2 of 2				
	Privitera et al.	1 of 4	3 of 4			
	**TOTAL**	4 of 9 (44%)	4 of 7 (57%)	1 of 3 (33%)	1 of 3 (33%)	
**Secukinumab + Vedolizumab**	Privitera et al.	2 of 2				
	**TOTAL**	2 of 2 (100%)				
**Vedolizumab + Apremilast**	Privitera et al.	1 of 1				1 of 1
	**TOTAL**	1 of 1 (100%)				1 of 1 (100%)
**Vedolizumab + Tofacitinib**	Glassner et al.			8 of 12		
	**TOTAL**			8 of 12 (67%)		

**Table 3 jcm-11-01076-t003:** Most used combinations.

Study	VEDO+ USTE	AntiTNF+ VEDO	AntiTNF+ USTE	TOFA+ VEDO	TOFA+ USTE	TOFA+ TNF	Other **
Goessens et al. * [10]	16	36	8	12	-	1	8
Glassner et al. [14]	25	7		8	3	9	1
Kwapisz et al. [12]	5	8	2				
Privitera et al. ** [13]	3	6	4				3
Yang et al. [11]	8	13	3				
TOTAL	62	75	20	21	3	10	19

VEDO (vedolizumab), USTE (ustekinumab), TOFA (tofacitinib). * CoTs used for extraintestinal manifestations were excluded. ** Other molecules used: apremilast, cyclosporine, rituximab, secukinumab, leflunomide, and tacrolimus.

## Data Availability

Not applicable.

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
