# Peer review of "Selecting the Best Combined Biological Therapy for Refractory Inflammatory Bowel Disease Patients"

_jcm, 2022, doi:10.3390/jcm11041076_

Round 1

Reviewer 1 Report

Very interesting and short review on dual biologic therapy in aggressive IBD. I like the review and it was easy to read and a nice update! 

This is not a new topic, it is not original but it provides a nice summary of the papers published on dual biologic therapy I’m aggressive IBD.

Minor improvements:  

introduction: -2nd paragraph: Need to discuss more on why a drug would be withdrawn and replaced by another. Give specific examples for example, mechanistic failure of anti TNF so replace with intern inhibitor.   

Discussion: -Please include a line or cite the paper by Molly stone published in Am J gastroenterology “the role of dual biologic therapy in IBD"

Author Response

Thank you for your review and your comments.

  • Introduction was modified and the withdrawn and replacement of the treatment was explained better with some examples
  • The article of Molly Stone was added in the discussion.

Reviewer 2 Report

Dear Author

  This article reviews an important issue of combine therapy for refractory IBD. The followings are my comments. 

1. Table 1, please provide more information about the study population, i.e. % of  CD and % of UC , patient age, and disease duration information from  each study if possible. 

2.  Line 85, should be UC not CU

3.  As mentioned, combination therapy for IBD may be prescribed for active IBD or active EIM without active IBD. The tittle of the article is "refractory IBD" and if so, please exclude patients with combination therapy for active EIM but not active IBD in the analysis.  For example, 23 patients have only active EMR received co-therapy among the 98 patients  in your reference 10. 

Author Response

Thank you for your review and your comments.

  • Table 1 was modified including the required information
  • Line 85 was changed, UC for CU
  • Regarding the mentioned study, patients with EIM were excluded from the analysis of clinical or endoscopic response/remission. However, data of safety were expressed globally and cannot be excluded. We added in the limitations and we modified the table 3 excluding the combinations for EIM

We attach the modified tables: 

 Reference

Year

Study type

Number of subjects

Disease

Age (mean)

Disease duration (mean years)

Clinical evaluation

Endoscopic evaluation

Adverse events

Follow up (mean)

Goessens et al. (10)

2021

Multicentric

Retrospective

98

58 CD

40 UC

26

70% response

50% response

42%

8 month

Glassner et al. (14)

2020

Unicentric

Retrospective

50

32 CD

18 UC

1 IBD-U

36.7

14.8

50% remission

34% remission

16%

8 month

Kwapisz et al. (12)

2021

Unicentric

Retrospective

15

14 CD

1 UC

36

12.5

73% response

44% response

53%

24 month

Privitera et al. (13)

2020

Multicentric

Retrospective

16

11 CD

5 UC

38

10.5

100% response

18.8%

7 month

Yang et al. (11)

2020

Multicentric

Retrospective

22

22 CD

35

50% response

50% response

13%

9 month

Study

VEDO+

USTE

TNF+

VEDO

TNF+

USTE

Tofa+

VEDO

Tofa+

USTE

Tofa+

TNF

Other**

Goessens et al*. (10)

16

36

8

12

-

1

8

Glassner et al. (14)

25

7

8

3

9

1

Kwapisz et al. (12)

5

8

2

Privitera et al**. (13)

3

6

4

3

Yang et al. (11)

8

13

3

TOTAL

62

75

20

21

3

10

19

Round 2

Reviewer 2 Report

Dear Editor

  The authors responded  to the questions raised. I have no more questions.